# Regulation of MT1-MMP Activity through Its Association with ERMs

**DOI:** 10.3390/cells9020348

**Published:** 2020-02-03

**Authors:** Henar Suárez, Soraya López-Martín, Víctor Toribio, Moreno Zamai, M. Victoria Hernández-Riquer, Laura Genís, Alicia G. Arroyo, María Yáñez-Mó

**Affiliations:** 1Molecular Biology Department, Universidad Autónoma de Madrid (UAM), 28049 Madrid, Spain; henar.suarez.montero@gmail.com (H.S.); victor.toribio@uam.es (V.T.); 2Severo Ochoa Molecular Biology Center (CBM-SO), Instituto de Investigación Sanitaria Princesa (IIS-IP), 28049 Madrid, Spain; slopez@cbm.csic.es; 3Unit of Microscopy and Dynamic Imaging, Centro Nacional de Investigaciones Cardiovasculares (CNIC), 28029 Madrid, Spain; mzamai@cnic.es; 4Vascular Pathophysiology Department, Centro Nacional de Investigaciones Cardiovasculares (CNIC), 28029 Madrid, Spain; victoria.hdez@hotmail.com (M.V.H.-R.); lauragenismartin@gmail.com (L.G.); agarroyo@cib.csic.es (A.G.A.); 5Molecular Biomedicine Department, Centro de Investigaciones Biológicas Margarita Salas (CIB-CSIC), 28040 Madrid, Spain

**Keywords:** MT1-MMP, ERM, tetraspanin enriched-microdomains, extracellular vesicles

## Abstract

Membrane-bound proteases play a key role in biology by degrading matrix proteins or shedding adhesion receptors. MT1-MMP metalloproteinase is critical during cancer invasion, angiogenesis, and development. MT1-MMP activity is strictly regulated by internalization, recycling, autoprocessing but also through its incorporation into tetraspanin-enriched microdomains (TEMs), into invadopodia, or by its secretion on extracellular vesicles (EVs). We identified a juxtamembrane positively charged cluster responsible for the interaction of MT1-MMP with ERM (ezrin/radixin/moesin) cytoskeletal connectors in breast carcinoma cells. Linkage to ERMs regulates MT1-MMP subcellular distribution and internalization, but not its incorporation into extracellular vesicles. MT1-MMP association to ERMs and insertion into TEMs are independent phenomena, so that mutation of the ERM-binding motif in the cytoplasmic region of MT1-MMP does not preclude its association with the tetraspanin CD151, but impairs the accumulation and coalescence of CD151/MT1-MMP complexes at actin-rich structures. Conversely, gene deletion of CD151 does not impact on MT1-MMP colocalization with ERM molecules. At the plasma membrane MT1-MMP autoprocessing is severely dependent on ERM association and seems to be the dominant regulator of the enzyme collagenolytic activity. This newly characterized MT1-MMP/ERM association can thus be of relevance for tumor cell invasion.

## 1. Introduction

Extracellular matrix (ECM) hydrolysis and rupture is performed by a large group of matrix metalloproteinases (MMPs), which can be either secreted into the extracellular milieu or transmembrane proteins. MT1-MMP, (also called MMP-14) belongs to this last group of transmembrane matrix metalloproteinases. First described as a gelatinolytic enzyme [1], MT1-MMP has many other ECM substrates such as collagen type I, proteoglycan, fibronectin, vitronectin and laminin-1 as well as α_1_-proteinase inhibitor and α_2_-macroglobulin [2]. Other well-known MT1-MMP substrates are pro-MMP-2 and pro-MMP-13 that have to be cleaved by MT1-MMP to become active [3,4]. Since MT1-MMP is expressed at the cell surface, some transmembrane receptors are also prone to be shed by this enzyme, such as CD44 [5], syndecan [6], integrin αv [7], transglutaminase [8] or KISS1 [9] among others.

MT1-MMP has a profound relevance in angiogenesis and cancer [10,11]. Since evidence showed a strong correlation between MT1-MMP expression levels and tumour invasiveness [12,13,14,15,16], a lot of research has been focused in unravelling the mechanisms that control its activity. Due to the variability of substrates cleaved by this enzyme and the low expression levels needed for detecting a significant effect, MT1-MMP activity has to be tightly regulated. Different mechanisms control either its gene expression and synthesis, or its activity via endocytosis, homodimerization, inhibition by TIMP-2, autocatalytic processing, posttranslational modifications and localization in specialized cell structures [17]. Processing of two main substrates of MT1-MMP, collagen and pro-MMP2, requires dimerization of the protease [18] through its hemopexin domain. This dimerization probably immobilizes the protein, facilitating its action over the collagen triple helix. For MT1-MMP-dependent MMP-2 activation, TIMP2, which was firstly described as an inhibitor of MT1-MMP, is able to interact with a MT1-MMP molecule creating a platform for pro-MMP2 binding. When this ternary complex is formed, another molecule of MT1-MMP can dimerize with the first one, through the hemopexin or the cytoplasmic region, and cleaves MMP-2 pro-domain [19]. Dimerization is also necessary for other means of regulation of MT1-MMP catalytic activity such as autocatalysis that results in MT1-MMP cleavage of its own catalytic domain to become inactive [20]. This requires at least two molecules of MT1-MMP close enough to enable their intermolecular auto-processing, creating two inactive species: a transmembrane form of 44 KDa and a soluble fragment of 18 KDa. The hinge region, hemopexin domain, transmembrane portion and the cytosolic tail form the 44 KDa fragment; while the 18 KDa fragment comprises the catalytic domain of the enzyme, although no associated functions have been found for the latter. Regarding the 44 KDa form, published results are somewhat contradictory. Some authors describe the 44KDa-MT1 form as a negative regulator of the active MT1-MMP, as it could compete with unprocessed molecules of MT1-MMP for homodimerization, avoiding their active status [21]. Other studies, however, report that those cells expressing 44KDa-MT1 have a higher growth rate in 3D collagen matrices and an enhanced pro-MMP2 activation. These positive effects could be explained by a decrease in the rate of MT1-MMP endocytosis [22] simply by competition of the fully active MT1-MMP with the autoprocessed form. TIMP-2 also seems to be a negative regulator of this auto-processing mechanism [23] and for the transient activation of MT1-MMP at invadopodia [24].

MT1-MMP function is also regulated by its inclusion into tetraspanin-enriched microdomains (TEMs) on the plasma membrane. TEMs act as organizers of the plasma membrane, directing the local concentration, subcellular localization and intramolecular interactions of their associated molecules. TEMs are, therefore, involved in cell migration and invasion processes, cell fusion and adhesion as well as in intracellular molecular trafficking [25]. Some tetraspanins have been identified as regulators of MT1-MMP activity [26]. CD81 increases migration and invasive phenotype of melanoma cells by the induction of MT1-MMP expression through the AKT signalling pathway [27], while CD63 can control its expression levels by promoting its lysosomal degradation [28]. In endothelial cells, tetraspanin CD151 emerged as a link between the metalloprotease and the integrin α3β1 controlling both its activity and localization [29]. MT1-MMP also interacts with tetraspanin CD9, which exert a positive regulation of the enzyme in combination with other tetraspanins [30].

MT1-MMP expression levels at the cell surface determine its degradative capacity, for this reason cells put a lot of effort in controlling its delivery and internalization from the plasma membrane through different molecules involved in vesicular trafficking, such as TOM1L1 [31], KIF5B [32] or Bet1 [33]. MT1-MMP endocytosis is regulated by the phosphorylation of two different residues (Thr567 or Tyr573) [34,35], depending on the signalling pathway involved. Several studies demonstrate that MT1-MMP endocytosis can be mediated by either clathrin- or caveolae-dependent pathways [36,37,38]. MT1-MMP can be recycled back to the plasma membrane after being internalized. While polarized delivery from the Golgi is controlled by Rab8 [39], its recycling and distribution into invadopodia is controlled by flotilins, VAMP7 and Rab7 among others [34,40,41]. Experiments where MT1-MMP endocytosis or recycling was blocked enabled an increase in the amount of protease expressed on the cell surface, however, invasion and migration were disrupted [34,42], confirming the relevance of this cycle for the modulation of MT1-MMP activity. It has also been demonstrated that invadopodia actively secrete exosomes enriched in MT1-MMP, enhancing the invasion process [41,43,44].

MT1-MMP localization, endocytosis and recycling are controlled by its short cytoplasmic domain that contains multiple sequences for the interaction with other proteins. The µ2 subunit of adaptor protein 2 (AP2) binds to the LLY sequence for its internalization [3]. Recycling of the enzyme seems to be controlled by the DKV sequence, through its interaction with LIMK [38]. Binding of MT1-MMP to MTCBP-1 drives its localization at invadopodia [45]. The cytoplasmic domain can also be palmitoylated, being this post-translational modification essential for cell migration and clathrin-dependent internalization [46]. MT1-MMP also regulates Rac activity via its association to p130Cas [47]. The cytosolic region also contains three clusters of positively charged amino acids involved in the interaction with FIH-1 [48], or with radixin [49], although the functional consequences of this last interaction was not pursued. We hypothesized that these positively charged clusters could enable the interaction with ERM (ezrin, radixin, moesin) proteins and may have an impact in MT1-MMP subcellular distribution and activity.

ERM proteins act as molecular linkers between transmembrane proteins and the actin cytoskeleton and, therefore, they play an important role in the organization and stability of specialized plasma membrane domains [50]. Members of this family share functional and structural features including two specialized domains: the FERM domain at the N-terminal region, which mediates ERM interaction with transmembrane proteins, and the C-ERMAD at the C-terminal with F-actin-binding capacity [51]. ERMs are found in the cytosol in a closed conformation, requiring their activation and mobilization towards the plasma membrane a phosphorylation step that will expose the actin-binding site. ERM molecules interact with a wide variety of cell surface proteins and receptors, such as CD44 [52], ICAM3 [53] or tetraspanins [54] creating specialized signalling platforms [55]. Previous reports have shown some evidence that relate moesin with MT1-MMP expression levels in tumour cells [56], however, the molecular mechanisms involved are unknown.

## 2. Materials and Methods

### 2.1. Cells and Cell Transfection

MCF-7 human breast adenocarcinoma cell line, that does not express endogenous MT1-MMP, were cultured in DMEM (GE Healthcare, Chicago, IL, USA) supplemented with 10% foetal bovine serum (FBS; GE Healthcare) and 0.01 mg/mL of insulin (Actrapid, Novo Nordisk, Bagsværd, Denmark). SUM159 human breast carcinoma cell line was cultured in Dulbecco’s modified Eagle’s medium (DMEM) F12 supplemented with 5% FBS (GE Healthcare), non-essential aminoacids (Gibco^TM^, Thermo Fisher, Waltham, MA, USA), 5 µg/mL of insulin (Actrapid, Novo Nordisk) and 1 µg/mL hydrocortisone.

MCF-7 cells (5 × 10^6^) were washed with PBS, and electroporated with 20 µg of DNA in OPTIMEM (Gibco, Invitrogen, Carlsbad, CA, USA) at 150 V and 950 µF (Gene Pulser II, Bio-Rad, Hercules, CA, USA). For selection of stable transfectants, cell culture medium was supplemented with 1 mg/mL of G418 and MT1-MMP expressing populations were obtained using a FACSARIA FUSION Cell Sorter (BD Biosciences, San Jose, CA, USA).

For CRISPR/Cas9 deletion of CD151 the pX330 plasmid that encodes for both GFP, Cas9 protein and presents a restriction site for the Bsb1 enzyme in which the guide RNA sequence is cloned, was employed. CD151 tetraspanin guides were directed against exon 1:

Guide RNA I

FW 5′ CACCGAGCAATTGTAGGTAAACAGC 3′

REV 5′ AAACGGCTGTTTACCTACAATTGCTC 3′

Guide RNA II

FW 5′ CACCGCTGGATGAAGGTCTCCAACT 3′

REV 5′ AAACAGTTGGAGACCTTCATCCAGC 3′

Transfected cells were cultured for 7 days, stained with anti-CD151 mAb and negatively selected in a FACSARIA FUSION Cell Sorter (BD Biosciences).

### 2.2. Antibodies and Reagents

Primary antibodies used were: anti-MT1-MMP (LEM-2/15) [57], anti-GFP (full-length polyclonal antibody; Living Colors, 632592), anti-ERM (90.3 rabbit polyclonal, kindly provided by Dr. H Furthmayr, Stanford, CA, USA), anti-phospho-ERM (Cell Signaling, Danvers, MA, USA), anti-EEA1 (N-19 polyclonal antibody, Santa Cruz, CA, USA), anti-LAMP1 (H4A3, Abcam, Cambridge, UK), anti-CD63 (Tea3/10) [58], anti-human CD151 (LIA1/1 or 11B1) [58].

### 2.3. Constructs and Reagents

Mutations in the cytosolic tail were done on the wt-MT1-MMP coupled to the mEGFP construct (tag at the C-terminal domain) [37] using QuikChange Lightning Site-Directed Mutagenesis Kit (Agilent Technologies, Santa Clara, CA, USA). Primers used for these mutations were: MT1-MMP-RRH563AAA fw: *CTTGCAGTCTTCTTCTTCG CAGCCGCTGGGACCCCCAGGCGAC* and rv: *GTCGCCTGGGGGTCCCAGCGGCTGCGAAGAAGAAGACTGCAAG*; MT1-MMP-RR569AA fw: *GACGCCATGGGACCCCCGCGGCACTGCTCTACTGCCAGCG* and rv: *CGCTGGCAGTAGAGCAGTGC CGCGGGGGTCCCATGGCGTC*; MT1-MMP-R576A fw: *CGACTGCTCTACTGCCAGGCTTCCCTGCTGGACAAGG* and rv: *CCTTGTCCAGCAGGGAAGCCTGGCAGTAGAGCAGTCG*.

mRuby-CD151 (tag at the N-terminal domain) has been previously described [59].

### 2.4. Enzyme-Linked Immunosorbent Assay (ELISA) In Vitro Binding Assay

Streptavidin binding plates (96 wells, Pierce) were incubated at 4 °C for 3 h in 0.1 M Na_2_CO_3_ buffer (pH 9.6) containing 40 µM of biotinylated peptides (GenScript, Piscataway, NJ, USA) corresponding to the MT1-MMP cytosolic tails: MT1-MMP (563RRHGTPRRLLYCQRSLLDKV582), MT1MMP-RRH563AAA, MT1-MMP-RR569AA and MT1-MMP-R576A. To avoid non-specific binding to biotinylated peptides, samples were blocked with 2% BSA/TBS-Tween containing 10% FBS overnight at 4 °C. GST-tagged recombinant N-moesin [60] was added to the wells in equal concentrations and incubated at room temperature (RT) for 1 h. After extensive washing, GST protein binding to the cytosolic tail sequences was detected using anti-GST ab and HRP-based detection. Optical density of an empty well was subtracted from all the absorbance data obtained.

### 2.5. Co-Immunoprecipitation, Internalization and Immunoblot Assays

For co-immunoprecipitation, stable transfectants on MCF-7 or CRISPR/Cas9 CD151 MCF-7 cells were lysed in 1% Brij96 in TBS with protease and phosphatase inhibitor cocktails (Roche). Lysates were precipitated with anti-GFP. Immunoprecipitates were washed 6 times with lysis buffer and resolved in 10% sodium dodecyl sulphate polyacrylamide gel electrophoresis (SDS-PAGE) under non-reducing conditions (for phospho-ERM or CD151 blotting) or reducing conditions (for loading control with anti-GFP).

For immunoblotting of total cell lysates, cells were lysed in PBS 1% Triton X-100 containing protease and phosphatase inhibitors (Roche). Lysates were analysed by 8% SDS-PAGE under reducing conditions.

Blots were revealed with FUJIFILM LAS-4000 after membrane incubation with specific antibodies and peroxidase-conjugated secondary antibodies (Pierce, Waltham, MA, USA).

For internalization follow-up, the same number of cells was seeded in multiwell plates and after 24 h they were washed with cold HBSS and labelled with biotin using 0.5 mg/mL EZ-Link sulfo-NHS-SS-biotin (sulfosuccinimidyl-20 (biotinamide) ethyl-1,3-dithiopropionate) (Invitrogen) for 30 min at 4 °C. Cells were washed twice with complete medium to remove unbound biotin and incubated for 0, 15 or 30 min at 37 °C and 5% CO_2_ to allow internalization of labelled proteins. Cells were then washed again with cold HBSS and incubated 15 min at 4 °C with 100 mM 2-mercaptoethane sulfonic acid (MESNA; Sigma, St. Louis, MO, USA) diluted in 50 mM Tris-HCl (pH 8.6), 100 mM NaCl, 1 mM EDTA, and 0.2% BSA. Cultures were washed again with cold PBS and incubated with 120 mM iodoacetamide (Sigma) for 10 min at 4 °C. The samples were lysed in 1% Triton X-100 in PBS with protease inhibitors (Roche, Basel, Switzerland). The lysates were incubated with sepharose-G protein beads, previously coupled to the anti-GFP antibody, overnight. After extensive washing, the samples were eluted with Laemmli buffer at 70 °C for 10 min and loaded into 8% SDS-PAGE gels under non-reducing conditions. The membranes were revealed with streptavidin-HRP (Invitrogen) using a Fujifilm-LAS 4000. After a stripping process, membranes were reblotted with anti-GFP for loading control.

### 2.6. Fluorescence Confocal Microscopy, Fluorescent Lifetime Imaging-Förster Energy Transfer (FLIM-FRET) and Flow Cytometry

Cells were plated onto 50 µg/mL of collagen (Nutragen, CellSystems, Troisdorf, Germany) for 6 h at 37 °C, fixed in 4% paraformaldehyde for 15 min, permeabilised with TBS 1%Triton X-100 for 5 min, stained with primary antibodies before secondary antibodies coupled to Alexa fluorochromes, and mounted using Fluoromount-G (SouthernBiotech, Birmingham, AL, USA). Images were acquired using a Leica TCS-SP5 confocal laser-scanning system equipped with Ar and He/Ne lasers and a Leica DMIRBE inverted microscope (Leica Microsystems, Heidelberg, Germany). For colocalization analysis, the entire z-stack was processed using Image J (NIH) software and the Coloc 2 plugin.

To quantitate Förster Energy Transfer (FRET) through by Fluorescent Lifetime Imaging (FLIM) [61], SUM-159 cells were transiently co-transfected with the different constructs of the MT1-MMP protein coupled to mEGFP and CD151 coupled to the Ruby fluorescent protein. 24 h after transfection, the cells were seeded on collagen-coated (20 µg/mL) MatTek plates. FLIM images were obtained in a scanning ALBA system coupled to a Nikon Ti-E inverted microscope (Nikon Corp., Tokyo, Japan) and equipped with a two-channel fast-FLIM card digital frequency domain (DFD) (ISS Inc., Champaign, IL, USA) with two H7422 photomultipliers (Hamamatsu Photonics), a Nikon MRD07600 CFI 60X/1.2-NA WI apochromatic Plan lens, incubator and a heating chamber (Okolab SRL). Two-photon excitation at 850 nm was obtained with a MaiTai DeepSee laser (Newport Corp., Irvine, CA, USA) and the emission was then collected through FF01-680/SP and BP530/43 filters (Semrock Inc., Idex Corporation, Lake Forest, IL, USA). The changes of fluorescence lifetime decay of the donor fluorophore (MT1-MMP-mEGFP) were studied by the Phasor-FLIM approach. Phasor-FLIM analysis was carried out using the SimFCS software (Globals Unlimited, LFD, Irvine, CA, USA) [61,62].

For flow cytometry, cells were lifted by Trypsin/EDTA treatment. When indicated, samples were stained with anti-MT1-MMP LEM2/15 or anti-CD151 LIA1/1 mAbs, followed by anti-mouse APC and analysed in a FACSCanto cytometer (BD Biosciences, San Jose, CA, USA) using the FlowJo software (version Tree Star, Ashland, OR, USA).

### 2.7. Exosome Isolation and Quantification

Cells were cultured in DMEM supplemented with 10% exosome-depleted FBS. After 6 days, supernatants were recovered and centrifuged first, at 400× *g* for 5 min and at 2000× *g* for 10 min to remove cells and cell debris. The cleared supernatant (15 mL) was concentrated by ultrafiltration 30 min at 2000× *g* using Amicon Ultra-15 Centrifugal Filter Units (Millipore, Billerica, MA, USA). The final volume of 0.2 mL was loaded onto a SEC column for extracellular vesicle (EV) purification as previously described [63]. Fractions enriched in EVs were detected by dot-blot, for that, 3 µL of each fraction were loaded onto a nitrocellulose membrane (0.22 µm GE Healthcare Life Sciences) and immunoblotted for anti-CD63 antibody. Only those three fractions with highest intensity values (commonly 6th-8th) were pooled. Protein concentration was measured using a BCA assay (Pierce, Thermo Fischer Scientific). Due to differences in protein concentration between samples, EVs were centrifuged at 100,000× *g* at 4 °C for 4 h and resuspended in an appropriate volume of PBS.

A modification of our bead-assisted flow cytometry assay [64,65], the ExoStep kit (Immunostep), was used to quantitate MT1-MMP incorporation into EVs. This assay is based on the capture of EVs on magnetic beads coated with an anti-CD63 antibody and staining with anti-CD9 antibody, since both CD63 and CD9 tetraspanins are highly enriched on the surface of EVs from most cell types. MT1-MMP sorting into EVs could be followed by the detection of the mEGFP fluorescence signal, while the CD9 signal allowed to normalize for EV content. For that, EVs were coupled to the beads overnight (ON) at RT, and stained with anti-CD9 biotinylated antibodies. Samples were analysed using a Gallios Cytometer (Becton Dickinson, Franklin Lakes, NJ, USA) and Kaluza Flow Cytometry Analysis (Beckman Coulter, Brea, CA, USA) or FlowJo softwares (Becton Dickinson, Ashland, OR, USA).

### 2.8. Extracellular Matrix (ECM) Degradation Assays

Gelatin-Rhodamine coated coverslips were prepared as previously described [66]. 70,000 cells were cultured on the coverslips for 6 h, fixed with 4% paraformaldehyde for 10 min and washed three times with TBS. Coverslips were mounted in Fluoromont-G medium (Southern Biotech, Birmingham, AL, USA). Confocal images were obtained with a Leica TCS-SP5. The degradation area was measured using Image J (NIH, University of Wisconsin, Madison, WI, USA) software.

### 2.9. Statistical Analyses

Statistical analyses were performed using GraphPad Prism (GraphPad Software Inc., San Diego, CA, USA). Normality test were performed and then P values were calculated using one-way analysis of variance (ANOVA) with Tukey’s post hoc multiple comparison test or Dunn’s when indicated. Statistical significance was assigned at * *p* < 0.05, ** *p* < 0.01, *** *p* < 0.001.

## 3. Results

### 3.1. MT1-MMP Interacts with ERM (Ezrin, Radixin, Moesin) Proteins through Basic Residues in Its Cytoplasmic Tail

ERM (ezrin, radixin, moesin) proteins act as molecular linkers by binding to both certain transmembrane proteins and the actin cytoskeleton. The cytoplasmic tail of MT1-MMP has three different clusters of positively charged amino acids, which is a common feature in proteins that establish interactions with ERM proteins [67]. To assess whether this is the case for MT1-MMP, we performed an enzyme-linked immunosorbent assay (ELISA) in vitro binding assay using synthetic peptides encoding the C-terminal sequence of MT1-MMP and the recombinant N-terminal domain of moesin fused to GST. In addition, each basic cluster in MT1-MMP cytosolic sequence was replaced by alanines. Our results demonstrated the interaction between wildtype (wt) MT1-MMP and moesin in vitro, that was completely abrogated by mutation of the juxtamembrane RRH563 cluster (Figure 1A). Mutation of the RR569 cluster also reduced the interaction, while mutation to alanine of the arginine in position 576 did not impair the binding (Figure 1A).

Complementarily, those same residues on a mEGFP-tagged MT1-MMP construct were mutated by site directed mutagenesis, and the different mutants transfected into the breast carcinoma cancer cell line MCF-7, which does not express endogenous MT1-MMP. Confocal microscopy analysis of these mEGFP-tagged versions of MT1-MMP revealed the colocalization with the phosphorylated (active) form of ERMs (Figure 1B), with a significant decrease in Pearson coefficient values for the RRH563AAA mutant (Figure 1B).

### 3.2. Mutation of the ERM Binding Site of MT1-MMP Alters the Steady State Subcellular Distribution of the Protein

Fluorescence microscopy analyses also suggested that RRH563AAA mutant was more prevalently located in intracellular vesicles, as compared with wild-type MT1-MMP. Therefore, we decided to analyse MT1-MMP distribution in different intracellular compartments. The RRH563AAA mutant showed the greater degree of colocalization with EEA-1, a marker for early endosomes, and also with CD63 or LAMP-1, markers of late endosomes and lysosomes (Figure 2A and Appendix A).

We thus decided to analyse the dynamics of internalization of MT1-MMP and the RRH563AAA mutant by labelling the surface proteins with biotin and following their internalization in time. The amount of internalized MT1-MMP was evaluated by Western blotting (Figure 2B). Despite its predominant presence in endosomal and lysosomal compartments at steady state, the internalization of the RRH563AAA mutant was slower than wt MT1-MMP.

After internalization, MT1-MMP might be incorporated in extracellular vesicles, since MT1-MMP secretion onto exosomes has been reported to be fundamental for cell invasion of 3D matrices [43]. Moreover, ERM molecules are highly abundant in EVs. We thus wondered whether the molecular association with ERMs would be relevant for MT1-MMP sorting into EVs. To analyse the incorporation of MT1-MMP into EVs, EVs were isolated from conditioned media and purified using size exclusion chromatography to eliminate possible protein contaminants. As shown in Figure 2C, faint mEGFP signal could be detected in EVs derived from conditioned media of the different MCF-7 cell cultures. There was not a significant difference in the presence in EVs between the wild type form of MT1-MMP and the RRH563AAA mutant (Figure 2C).

### 3.3. Effect of ERM Binding on MT1-MMP Distribution in Membrane Microdomains

We next decided to focus on the MT1-MMP population at the plasma membrane responsible for its activity. It has been previously described that tetraspanin CD151 directly associates with MT1-MMP and regulates its ECM degradation activity by establishing a ternary complex with the protease and the integrin alpha3beta1. On the other hand, ERM molecules are also directly connected to tetraspanin-enriched microdomains (TEMs) [54]. We thus decided to analyse whether the mutations in the ERM binding motifs of MT1-MMP cytoplasmic region could affect its association with the tetraspanin CD151 by FLIM-FRET analysis in cells co-transfected with the different mEGFP-tagged MT1-MMP versions (wild-type and its mutants) together with a CD151-Ruby construct (Figure 3A and Methods). The phasor-FLIM analyses revealed that all MT1-MMP-mEGFP tagged versions showed a significant FRET efficiency at the plasma membrane, suggestive of interaction with CD151, clearly different form the fluorescence lifetime associated to no-FRET signal in cells in which only the MT1-MMP-mEGFP construct, without the CD151-Ruby acceptor, was transfected. In addition, FRET was higher in cell protrusions and membrane spikes expressing wt MT1-MMP-mEGFP or the R576A mutant together with CD151-Ruby, while these high FRET areas were reduced in cells expressing the RR569AA mutant and almost completely abrogated in cell expressing the RRH563AAA mutation (Figure 3B). These data indicate that the mutation of these basic clusters in MT1-MMP cytosolic region does not impede the association of the metalloproteinase with CD151 but impairs the accumulation and coalescence of MT1-MMP-contaning TEMs at actin-rich structures (Figure 3A).

Therefore, mutation of the ERM binding site did not seem to affect inclusion of MT1-MMP into TEMs. To corroborate that these molecular associations are independent of each other, we generated MCF-7 cells defective for CD151 by using the CRISPR/Cas9 gene deletion system. We confirmed the absence of CD151 by flow cytometry (Figure 4A) and generated mEGFP-tagged stable transfected cell lines of the different versions of MT1-MMP on this CD151-KO background. In these cells, we observed the same pattern of colocalization with phospho-ERM proteins of the wt MT1-MMP and the juxtamembrane RRH563AAA mutant in the absence of CD151 (Figure 4B). In addition, a similar subcellular distribution was observed in both wt MCF-7 and CD151 KO cells, where RRH563AAA accumulated in late endosomes and lysosomes (Figure 4C and Appendix A). To directly assess whether the association of MT1-MMP with ERMs and CD151 are independent phenomena, we performed co-immunoprecipitation analyses in detergent conditions that maintain the integrity of tetraspanin-eriched microdomains (Figure 4D). In these assays, we could detect the co-immunoprecipitation of MT1-MMP with active-ERM in both, MCF-7 and CD151-KO cells, which was in both cases disrupted by mutation of the RRH563 motif to AAA. Moreover, both MT1-MMP and the RRH563AAA were able to co-immunoprecipitate CD151 in MCF-7 cells. Loading control of anti-GFP immunoprecipitates was performed in a parallel SDS-PAGE gel under reducing conditions. We corroborated that expression of the different MT1-MMP constructs or CRISPR/Cas9 deletion of CD151did not affect the levels of active ERM in the cells by western blot analyses on whole cell lysates (WCL) obtained from the different cell lines (Figure 4D).

### 3.4. Effect of ERM Association on MT1-MMP Autoprocessing and Activity

Our data suggest that ERM connection controls MT1-MMP lifetime at the plasma membrane as well as its coalescence in big clusters at cell protrusions. Therefore, we decided to quantify the relative expression of MT1-MMP at the plasma membrane by flow cytometry after staining non-permeabilized cells with an anti-MT1-MMP LEM-2/15 mAb. In these analyses, we could observe that, indeed, the membrane expression levels of the RRH563AAA mutant were around 1.5 folds those of the wt construct (Figure 5A).

Since autocatalytic cleavage of MT1-MMP catalytic domain is a major regulatory mechanism of MT1-MMP activity that will be in principle regulated by MT1-MMP clustering, we studied the effect of mutations in the MT1-MMP cytosolic domain on its auto-processing. For that purpose, whole-cell lysates of cells transfected with the different mutants were analysed by western-blot with anti-GFP antibody. These assays revealed a clear effect of ERM association so that the RRH563AAA mutant showed a lower degree of autocatalytic processing, while no difference was observed with other cytoplasmic region mutants (Figure 5B, quantitated in stable transfectants of wt and RRH563AAA mutants in Figure 5E).

To assess the impact of ERM binding on MT1-MMP enzymatic activity directly, cells were plated onto gelatin-rhodamine for ECM degradation assays. Cells transiently transfected with MT1-MMP mutants tagged with mEGFP were analysed for their motility and matrix proteolytic potential. No major differences were observed in random migration rate among the different mutants (not shown). In these experiments, gelatinolytic activity was found to be completely dependent on MT1-MMP expression, since those cells showing no GFP signal did not show any associated degradation area (Figure 5C). Importantly, as shown in Figure 5C, we could observe a significant increase in the gelatinolytic capacity of the RRH563AAA mutant compared to wt MT1-MMP.

To confirm that the effects observed on MT1-MMP autoprocessing and gelatinolytic activity were completely attributable to MT1-MMP association to ERMs, we analysed these processes in cells that lacked CD151 expression but stably expressed either wt MT1-MMP or the RRH563AAA mutant. We found that indeed both autoprocessing and overall gelatinolytic capacity were not affected by the absence of CD151 expression on MCF-7 cells but again altered in the RRH563AAA mutant in MCF-7 CRISPR/Cas9 CD151 cells (Figure 5D–F).

## 4. Discussion

The subcellular localization is extremely critical in modulating the activity of MT1-MMP. The regulation of intracellular trafficking, inclusion in extracellular vesicles as well as dimerization and inclusion in microdomains at the plasma membrane play a crucial role in the functional regulation of MT1-MMP [18]. Here, we aimed at exploring the putative connection of MT1-MMP with the cortical actin cytoskeleton via ERM linkers. Our data indicate that a series of basic amino acids in the cytosolic region of MT1-MMP are necessary for interaction with ERMs. In a previous crystallographic work, it was already described how the cytosolic domain of MT1-MMP could interact with radixin through the sequence 568PxøLLY and that the amino acid R576 was also relevant in stabilizing the association through hydrogen bonds [49]. In our mutants, the elimination of this residue did not seem to affect the interaction with ERMs, which probably indicates that it is not critical for the binding, but once the proteins are close enough, it could participate in the stabilization of the complex. In the previous crystallographic study, the authors used crystals of radixin bound to a peptide with the carboxyl terminal sequence of MT1-MMP. This peptide, not being anchored to a membrane, had a conformational flexibility that precluded the determination of the precise conformation of the juxtamembrane section. In our experimental set up, it is precisely the presence of amino-terminal juxtamembrane sequence the most important condition for the interaction.

To have a functional readout of the relevance of MT1-MMP interaction with ERMs, we used mutants that maintained the protein in its native state, in a breast carcinoma cell model that has no endogenous expression of MT1-MMP. In these MCF-7 cells, we were able to confirm that the RRH563 amino acid sequence is fundamental for the colocalization and co-immunoprecipitation of MT1-MMP with the phosphorylated active form of ERMs. By mutating the RRH motif and abolishing the interaction with ERMs, the metalloprotease subcellular distribution and internalization rate is affected. Colocalization analyses with early and late endosome markers showed that mutation of the RRH motif increased the location of MT1-MMP in late endosomes and lysosomes but, surprisingly, internalization rate was reduced. MT1-MMP has been shown to be internalized via either clathrin- or caveolin-dependent pathways [42,68]. MT1-MMP binds to the AP2 clathrin adaptor via de LLY internalization motif located at position 573, which is maintained in our mutants. Syntaxin-4 binding has also been reported to be crucial for transport of MT1-MMP to the plasma membrane and localization to invadopodia [69,70]. We could not observe any differences in MT1-MMP co-immunoprecipitation with syntaxin-4 after mutation of the RRH motif (data not shown). However, disruption of MT1-MMP association with ERMs alters its clustering at membrane protrusions and this could directly affect MT1-MMP internalization and recycling [39]. Due to the increase in localization of MT1-MMP in positive compartments for CD63, and since MT1-MMP is a component of exosomes in invasive cells [43,44,71], we postulated that ERM association could be relevant for inclusion of the metalloproteinase in these vesicles. However, we were not able to detect significant differences in the inclusion of RRH563AAA mutant MT1-MMP in comparison to wt MT1-MMP into exosomes in our cellular model.

The local concentration of MT1-MMP at the plasma membrane can also be regulated by its inclusion in tetraspanin-rich microdomains, which are also connected to the actin cytoskeleton by ERMs. However, our data on the FLIM-FRET experiments, colocalization and co-immunoprecipitation with active ERMs in CD151 KO cells indicated that the interaction with ERMs is not necessary for the interaction of the protease with CD151 and vice-versa. In MCF-7 cells, gene deletion of CD151 by CRISPR/Cas9 technology did not influence the colocalization pattern of the different mutants of MT1-MMP with ERMs, nor MT1-MMP enzymatic activity both on itself and on the extracellular matrix. In a previous report from our group, we demonstrated that, in endothelial cells, MT1-MMP forms a ternary complex with integrins extracellular matrix receptors via CD151 [29], that plays an important role in the degradation activity of the protease on the matrix. In contrast, in the tumour cell line used in this study, the expression levels of the integrin alpha3beta1 were almost undetectable by flow cytometry (data not shown). Moreover, gelatinolytic activity in MCF-7 cells seems to be completely dependent on MT1-MMP expression, while in endothelial cells, other metalloproteinases such as MMP-2 also play a role in gelatin degradation. In addition, in the absence of CD151, other tetraspanin members, that associate with MT1-MMP in a looser way [30] could mediate its insertion into microdomains. In this regard, MT1-MMP has been shown to be palmitoylated at Cys 574, being this lipidation relevant for clathrin internalization [46]. Tetraspanins, and many partners inserted in TEMs are also palmitoylated, so MT1-MMP palmitoylation may be sufficient for TEM insertion in the absence of CD151 expression.

Although at the molecular level MT1-MMP association with either ERMs or CD151 seems to be independent phenomena, the subcellular distribution of TEMs that included MT1-MMP was severely affected in the case of the RRH563AAA mutant. Cells that expressed the construction of wild-type MT1-MMP displayed very high FRET values in cellular protrusions, where there would be a very high concentration of both metalloprotease and tetraspanin thanks to the interaction of both with ERMs, while these areas of high FRET were completely absent for the RRH563AAA mutant. This clustering effect could also explain the decrease in autoproteolytic processing, which would require a sufficient local concentration of the enzyme on the cell surface that facilitate two molecules of MT1-MMP to dimerize and cut. MT1-MMP dimerization has been shown to be regulated by soluble factors as chemokines in endothelial cells in which it requires an intact actin cytoskeleton [72]. The possible regulation of MT1-MMP/ERM association by intracellular signals or phosphorylation, therefore, deserves further studies. The combination of reduced internalization rate and reduced autoprocessing resulted in a slight increase in the membrane expression levels of the active protease. Our data by flow cytometry revealed that this was indeed the case. However, it should be taken into account that the LEM2/15 mAb used for flow cytometry does not recognized the autoprocessed form of the enzyme, since it is directed against the catalytic domain. Functionally, this autoproteolytic event seems to have a dominant effect on gelatinolysis activity in MCF-7 cells, so that in the case of the mutant of the RRH motif, self-processing was prevented and the net result was a greater proteolytic activity on the extracellular matrix. Since the dynamics of MT1-MMP turnover and insertion into the plasma membrane are critical for its pro-invasive actions [24], the increased activity of MT1-MMP in the absence of ERM association will probably directly impact also in a 3D scenario in tumour cell invasion. It has been recently reported that moesin knock-down impacts on MT1-MMP expression levels in oral cancer cells [56]. In this report, however, the autoprocessing of the protease or its direct interaction with moesin was not. In addition, a reduction in the expression levels of moesin may have additional effects due to its downstream signalling capacity, which may not be directly caused by the impairment of MT1-MMP connectivity to the cytoskeleton or its clustering at the plasma membrane.

Clustering of TEMs via ERM connection may not only regulate MT1-MMP dimerization and autoprocessing, but it could also play a role in regulating substrate accessibility, via connections with tetraspanin-associated molecules [73] or transmembrane receptor also linked to ERMs [52], such is the case of CD44, an important MT1-MMP substrate involved in cell migration [5,74]. In this regard, moesin is an adverse prognostic marker for several tumours, including breast cancer [75]. It is possible that interfering with MT1-MMP and ERM association may deregulate the pro-invasive actions of MT1-MMP in 3D invasion, by altering not only metalloproteinase action on the extracellular matrix but also by influencing the tumour cell membrane phenotype by regulating receptor shedding.

From the multiple works that exist to date, the general concept that emerges is that the activity of MT1-MMP on the cell surface depends on its dynamic internalization and on systems that maintain its stability on the surface. In our model, we could observe that blocking MT1-MMP interaction with ERMs delocalizes its subcellular localization at the membrane, which directly affected both its internalization dynamics and its autocatalytic processing, finally resulting in deregulated ECM degradation.

## Figures and Tables

**Figure 1 cells-09-00348-f001:**
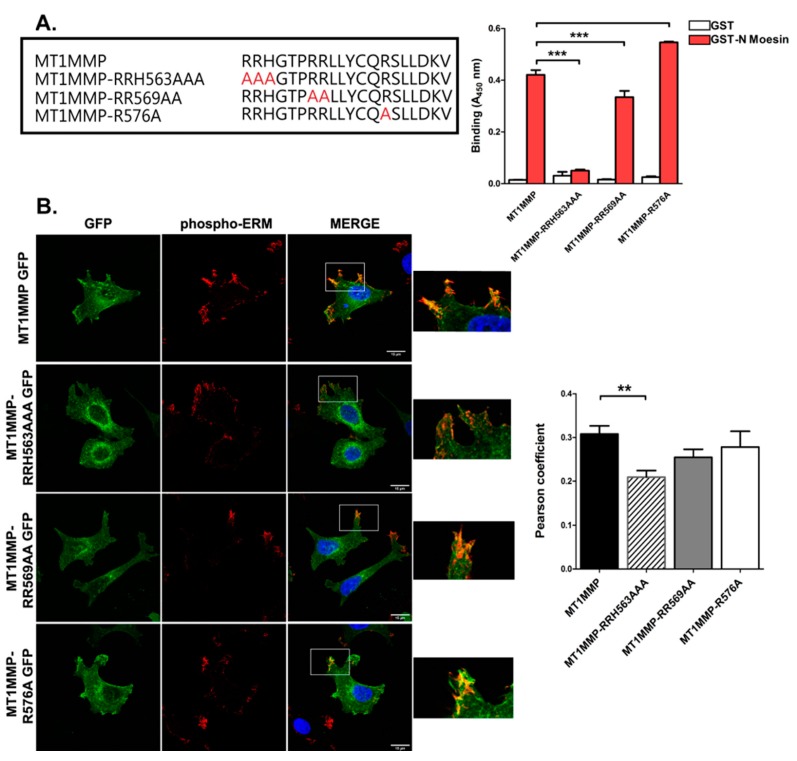
MT1-MMP cytoplasmic region interacts with ERMs (ezrin, radixin, moesin). (**A**) In vitro binding assays were performed using synthetic peptides encoding the wt C-terminal sequence of MT1-MMP or different mutant versions in which basic residues were exchanged for alanines (in red) and the recombinant N-terminal domain of moesin fused to GST. Data shown correspond to absorbance at 450 nm ± standard error of the mean (SEM) from three independent experiments, Statistical significance was determined using Tukey’s multiple comparison test, *** *p* < 0.001. (**B**) Confocal microscopy images of MCF-7 transiently transfected with MT1-MMP mEGFP and the different cytoplasmic mutants. Cells were stained for the endogenous active form of ERMs (phospho-ERM in red). Bars 15 µm. Graph represents the colocalization analysis between the green and the red channel. Data represent mean ± SEM of *n* ≥ 10 cells for each condition. Statistical significance was determined using Tukey’s multiple comparison test, ** *p* < 0.01.

**Figure 2 cells-09-00348-f002:**
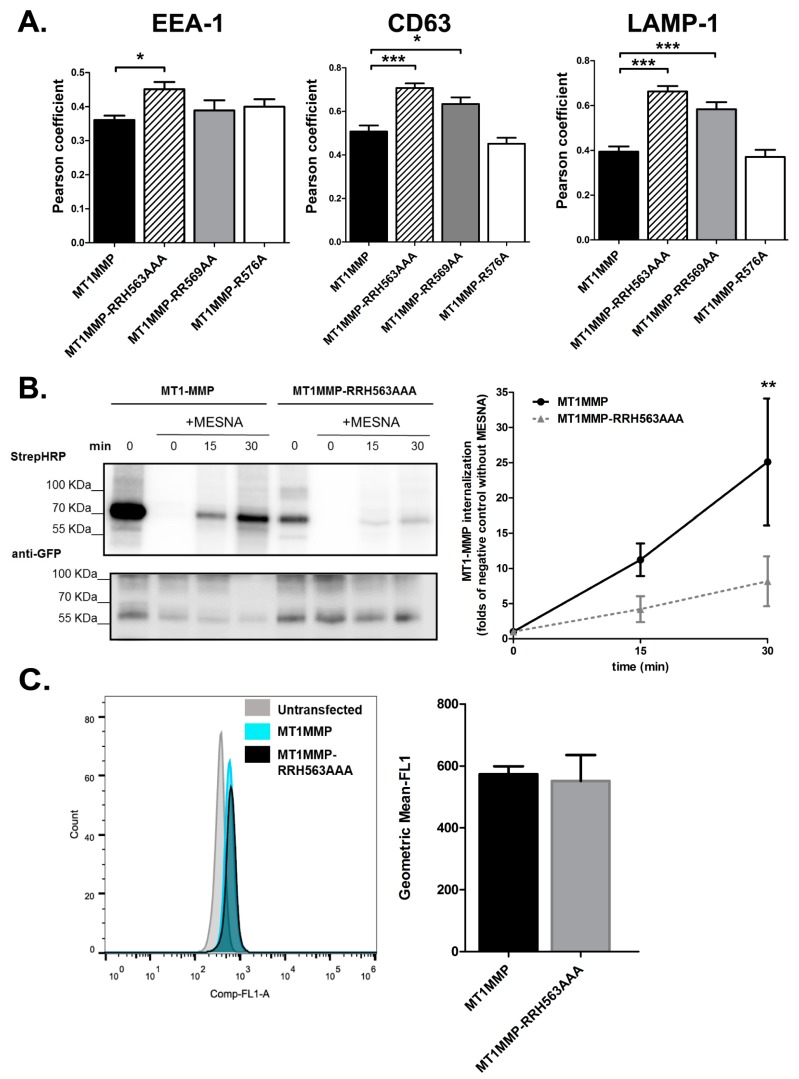
Mutation of the cytoplasmic RRH563 cluster regulates MT1-MMP internalization. (**A**) Confocal microscopy analyses of MT1-MMP wt or the RRH563AAA mutant colocalization with EEA-1, CD63 or LAMP-1, markers of early-, late-endosomes and lysosomes, respectively. Pearson coefficients were calculated for all mutants stained for EEA-1, CD63 or LAMP-1 *n* ≥ 10 cells per condition. Data represent mean ± SEM. p values were calculated using one way analysis of variance (ANOVA) with Tukey’s or Dunn’s post-hoc multiple comparison test when appropriate. * *p* < 0.05, *** *p* < 0.001. (**B**) The internalization of MT1-MMP was evaluated using a biotinylation assay. Gels show lysates of MCF-7 cells stably expressing MT1MMP-GFP or mutant MT1MMP-RRH563AAA GFP immunoprecipitated using the anti-GFP antibody after biotinylation of the cells and internalization of the biotinylated proteins, that was allowed for 0, 15 and 30 min before treating with mercaptoethane sulfonic acid (MESNA) and lysis. Control samples correspond to cells without MESNA treatment. Blotting was sequentially performed with streptavidin-HRP (StrepHRP) and anti-GFP. The graph represents the ratio of biotinylated MT1-MMP/total MT1-MMP related to that ratio in negative control sample without MESNA treatment ± SEM in three independent experiments; data were analysed by two-way ANOVA with Bonferroni’s post-test ** *p* < 0.01. (**C**). Bead-assisted flow cytometry analysis of GFP-tagged MT1-MMP and RRH563AAA mutant content in extracellular vesicles (EVs), derived from MCF-7 cells transfected with either wt or mutant MT1-MMP. Bar plot depicts the mean ± SEM of three independent experiments, they were analysed by two tailed unpaired Student *t*-test and no differences were detected between conditions.

**Figure 3 cells-09-00348-f003:**
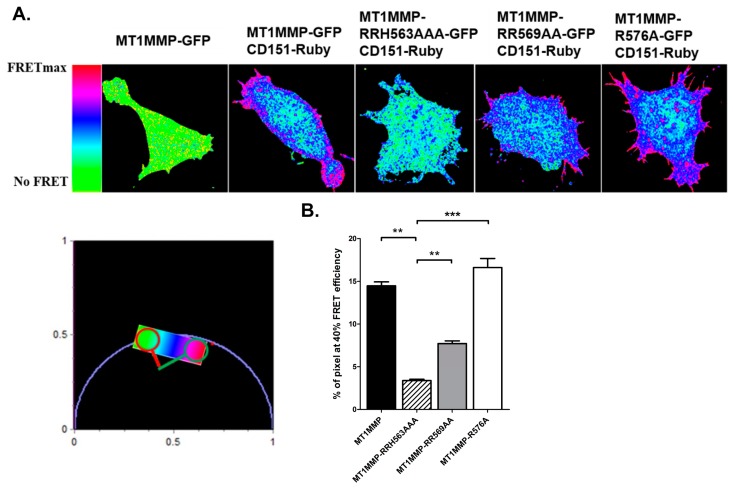
MT1-MMP connection to ERMs and tetraspanin CD151 are independent phenomena. (**A**) SUM159 cells were cotransfected with either wt-MT1-MMP-mEGFP or each of the three mEGFP-tagged mutants and CD151-Ruby. The decrease of the donor, mEGFP, fluorescence lifetime due to Förster energy transfer (FRET) was analysed by phasor analysis. The fluorescence lifetime distribution in donor-only cells is homogeneous at the cell membrane, and it sharply moves towards lower values as represented by the pseudo colour scale of the phasor plot. The related FRET efficiencies are reported also in pseudo colour scale as FRET efficiency images (**B**). Graph represents the percentage of pixels showing FRET efficiency over 40% analysed by fluorescent lifetime imaging (FLIM)-FRET in each of the mutants. Data are the mean ± SEM of *n* > 6 cells analysed by Kruskal-Wallis and Dunn′s multiple comparisons test ** *p* < 0.01, *** *p* < 0.001.

**Figure 4 cells-09-00348-f004:**
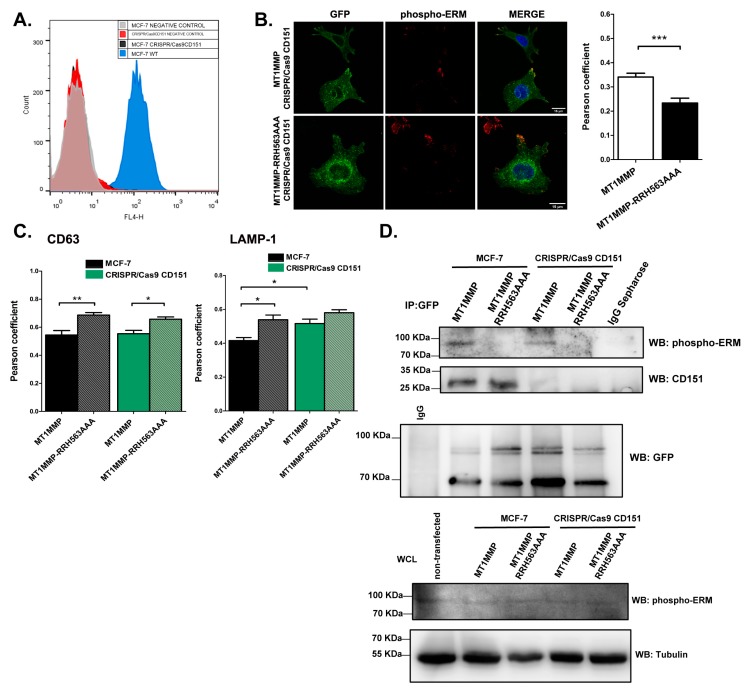
CD151 gene deletion does not impact on MT1-MMP subcellular localization. (**A**) Expression of tetraspanin CD151 was deleted using the CRISPR/Cas9 system in MCF-7 cells and analysed by flow cytometry to confirm the absence of the tetraspanin, after staining with LIA1/1 anti-CD151 mAb and an appropriated secondary antibody. Blue histogram corresponds to wt MCF-7 cells, black to CD151 knockout (KO) cells. Negative controls of unstained MCF-7 cells and unstained CD151 KO cells are shown in light grey and red histograms, respectively. (**B**) MCF-7 CD151 KO cells were transiently transfected with MT1-MMP mEGFP and the RRH563AAA cytoplasmic mutant. Cells were stained for the endogenous active form of ERMs (phospho-ERM in red). Colocalization analyses show a significant decrease in Pearson coefficient for the RRH563AAA mutant as compared with the wild-type form of MT1-MMP. Bars 15 µm. Graph represent the colocalization analysis between the green and the red channel. Data represent mean ± SEM of n≥10 cells for each condition. Statistical significance was determined using two tailed paired Student *t*-test, *** *p* < 0.001. (**C**) MCF-7 wt or CRISPR/Cas9 CD151 cells were transfected with mEGFP-tagged MT1-MMP constructs (wt or RRH563AAA) and plated onto 20 µg/mL collagen I-coated coverslips, fixed and labelled with anti-CD63 or anti-LAMP-1. Graphs show the colocalization analysis with CD63 wt (black) and MT1MMP-RRH563AAA mutant (grey) in MCF-7 cells and of wt (solid green) and MT1MMP-RRH563AAA mutant (light green) in CRISPR/Cas9 CD151 KO cells. The degree of colocalization was evaluated by means of the Pearson coefficient (PC) represented as the mean ± SEM of more than 15 cells of 3 independent experiments and analysed using one-way ANOVA and Dunn′s multiple comparisons test * *p* < 0.05, ** *p* < 0.01. (**D**) The different MCF-7 cell lines were lysed in 1% Bri96 and immunoprecipitated with anti-GFP polyclonal Ab or IgG sepharose. Samples were run under non-reducing conditions and blotted with anti-phospho ERM or anti-CD151 (11B1) Abs. For loading control, parallel samples were resolved under reducing conditions and blotted with anti-GFP Ab. Whole cell lysates (WCL) of MCF-7 cells and CD151 KO cells stably expressing MT1-MMP wt and MT1MMP-RRH563AAA mutant were immunoblotted for phospho-ERM. Tubulin was used as loading control. No differences were found among the different cell lines.

**Figure 5 cells-09-00348-f005:**
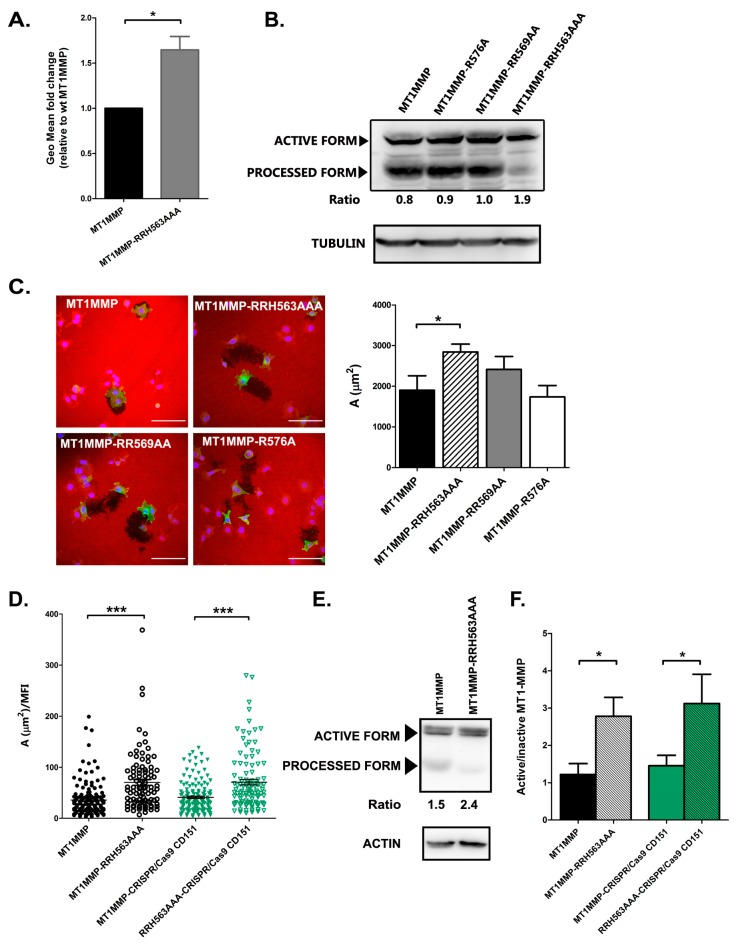
Connection to ERMs regulate MT1-MMP autoprocessing and activity. (**A**) Flow cytometry analysis of MT1-MMP surface expression represented as the mean fold change ± SEM of the fluorescence geometric mean comparing to MT1-MMP in three independent experiments, * *p* < 0.05 in a two-tailed paired Student *t*-test. (**B**) Western blot revealed with anti-GFP antibody was used to analyse the relative abundance of autoprocessed MT1-MMP form in total cell lysates. The ratio between full active molecule versus the inactive autoprocessed form is depicted below. Tubulin was used as loading control. (**C**) A fluorescent gelatin matrix was used as a substrate to measure MT1-MMP activity. MCF-7 cells were transiently transfected with the different mEGFP-tagged MT1-MMP constructions and plated onto 2 mg/mL gelatin for 6 h, fixed and the degraded area was quantified. Bars 100 µm. Data represent mean ± SEM of *n* ≥ 25 cells. Data was by Kruskal–Wallis test, * *p* < 0.05. (**D**) MCF-7 wt or CRISPR/Cas9 CD151 cells were transfected with mEGFP-tagged MT1-MMP constructs (wt or RRH563AAA) and plated onto 2 mg/mL fluorescent gelatin for 6 h before fixation. Degradation area was measure with Image J and normalized to the Mean Fluorescent mEGFP Intensity of each cell. Data are represented as the mean ± SEM of 100–150 cells from 3 independent experiments and by one-way ANOVA with Dunn’s post-test *** *p* < 0.001. (**E**) Autocatalytic cleavage reduction is also detected when the RRH563AAA mutant is transfected in CRISPR/Cas9 CD151 cells. Western blot with anti-GFP antibody was used to analyse the expression of MT1-MMP (wt or RRH563AAA) in MCF-7 CRISPR/Cas9 CD151 cells. The ratio between the active enzyme versus the processed form is calculated. Actin was used as loading control. (**F**) Graph depicts the autocatalytic activity of MT1-MMP as the ratio between the active enzyme versus the processed form obtained by western-blot. Data represent the mean ± SEM of five independent experiments analysed by one way ANOVA and Bonferroni’s multiple comparison test, * *p* < 0.05.

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
