# Peer review of "Regulation of MT1-MMP Activity through Its Association with ERMs"

_cells, 2020, doi:10.3390/cells9020348_

Round 1
Reviewer 1 Report
The manuscript investigates the regulation of the MT1-MMP membrane-bound metalloprotease by moesin, which is a member of the ERM family of intracellular proteins that links the cytoplasmic tails of membrane proteins with the actin cytoskeleton. MT1-MMP is of major research interest because of its role in cancer, with approximately 100 papers published in each of the last few years. ERM proteins are the focus of a comparable number of papers. The current manuscript follows on from a paper from a different group, which showed that moesin interacts with MT1-MMP and reported the crystal structure of the moesin FERM domain bound to the MT1-MMP cytoplasmic tail (Terawaki 2015 Genes Cells).
The current manuscript demonstrates the importance of three basic residues in the MT1-MMP tail (RRH) for binding to moesin. A mutant form of MT1-MMP that cannot bind moesin is then compared to wild-type MT1-MMP in a relatively simple breast cancer cell line system, MCF7, which lacks endogenous MT1-MMP. The mutant was found to be somewhat mis-localised in terms of its reduced co-localisation with active ERM proteins, partially impaired internalisation from the plasma membrane and increased localisation to endosomes and lysosomes. FRET and co-immunoprecipitation were used to show that the MT1-MMP interaction with active ERM proteins is independent from the MT1-MMP interaction with tetraspanin CD151, an interaction previously reported by the authors and another group. Finally, the mutant MT1-MMP was found to exist in a predominantly active, non-auto-processed form, unlike wild-type MT1-MMP, and to have increased MT1-MMP activity as measured by a gelatin degradation assay.
The findings in the manuscript extend our knowledge of MT1-MMP regulation and are likely to initiate new lines of research that aim to determine the importance of ERM proteins for MT1-MMP function in cancer. The cell line system is somewhat limited, but the range of techniques employed are impressive. I have the following comments and questions for the authors.
The Introduction is potentially not fully up-to-date for the relatively fast-moving MT1-MMP field. The Introduction has 40 references, but only two are from the last four years, despite over 400 MT1-MMP papers having been published in this period.
The Introduction does not cover ERM proteins very well, and is limited to just one sentence and no reference to describe them. A few more sentences and referencing would be useful to introduce what is a major focus of the manuscript.
Figure 1A. It would be useful to include statistical analyses of the bar chart data to support the authors’ conclusions.
Figure 2B. It would be useful to have less tightly cropped blots and to show the position of molecular weight markers above and below the bands; my molecular weight marker comment also applies to the other western blots in the manuscript. The data has been statistically analysed but appears to be n=2. It would be better to have at least n=3 for statistical analyses.
Figure 2C. The histogram data needs statistical analysis, particularly because the authors state in the text that the data is significant.
Figure 3A. Similar to the previous comment, the FRET data needs some quantitation and statistical analysis to support the contention in the text that the data is significant.
Figure 3C. A whole cell lysate blot of p-ERM would be useful to confirm that p-ERM levels are normal in the mutant transfectants.
Figure 4. It would be useful to clarify if this data is from stable or transient transfectants.
Figure 4A. What is the n number?
Figure 4B and 4E. The data is quantified but some statistical analyses of at least n=3 would be helpful, particularly as there appear to be differences in wild-type ratios between the two panels.
Figure 4C. What are the relative expression levels of wild-type versus mutant MT1-MMP and does the expression level affect the degree of gelatin degradation? It is importance to establish that this key data is not merely a result of different expression levels.
Figure S1. This would seem suitable as a main figure. In panel A, only one negative control histogram is shown, but there are two different cell types; it would be better to have both negatives. In panel B, there is a labelling error: CD9 should be CD151.
Author Response
Reviewer 1
The manuscript investigates the regulation of the MT1-MMP membrane-bound metalloprotease by moesin, which is a member of the ERM family of intracellular proteins that links the cytoplasmic tails of membrane proteins with the actin cytoskeleton. MT1-MMP is of major research interest because of its role in cancer, with approximately 100 papers published in each of the last few years. ERM proteins are the focus of a comparable number of papers. The current manuscript follows on from a paper from a different group, which showed that moesin interacts with MT1-MMP and reported the crystal structure of the moesin FERM domain bound to the MT1-MMP cytoplasmic tail (Terawaki 2015 Genes Cells).
The current manuscript demonstrates the importance of three basic residues in the MT1-MMP tail (RRH) for binding to moesin. A mutant form of MT1-MMP that cannot bind moesin is then compared to wild-type MT1-MMP in a relatively simple breast cancer cell line system, MCF7, which lacks endogenous MT1-MMP. The mutant was found to be somewhat mis-localised in terms of its reduced co-localisation with active ERM proteins, partially impaired internalisation from the plasma membrane and increased localisation to endosomes and lysosomes. FRET and co-immunoprecipitation were used to show that the MT1-MMP interaction with active ERM proteins is independent from the MT1-MMP interaction with tetraspanin CD151, an interaction previously reported by the authors and another group. Finally, the mutant MT1-MMP was found to exist in a predominantly active, non-auto-processed form, unlike wild-type MT1-MMP, and to have increased MT1-MMP activity as measured by a gelatin degradation assay.
The findings in the manuscript extend our knowledge of MT1-MMP regulation and are likely to initiate new lines of research that aim to determine the importance of ERM proteins for MT1-MMP function in cancer. The cell line system is somewhat limited, but the range of techniques employed are impressive. I have the following comments and questions for the authors.
1.- The Introduction is potentially not fully up-to-date for the relatively fast-moving MT1-MMP field. The Introduction has 40 references, but only two are from the last four years, despite over 400 MT1-MMP papers having been published in this period.
Thanks for this comment. We have updated the introduction following your suggestion focusing on those recent manuscripts that deal with MT1-MMP activity regulation (new refs 14-17, 31-33, and the accompanying text in the introduction section)
2.- The Introduction does not cover ERM proteins very well, and is limited to just one sentence and no reference to describe them. A few more sentences and referencing would be useful to introduce what is a major focus of the manuscript.
Following the reviewer request we have expanded the introduction devoted to ERM proteins (last paragraph of introduction and new references 50-56).
3.- Figure 1A. It would be useful to include statistical analyses of the bar chart data to support the authors’ conclusions.
Statistical analysis has been performed and included for data in Figure 1A.
Figure 2B. It would be useful to have less tightly cropped blots and to show the position of molecular weight markers above and below the bands; my molecular weight marker comment also applies to the other western blots in the manuscript. The data has been statistically analysed but appears to be n=2. It would be better to have at least n=3 for statistical analyses.
We now show less tightly cropped blots as well as the position of molecular weight markers around the bands. Molecular weight markers have been also added to gels in other Figures. We have included a third experiment to the quantitation.
Figure 2C. The histogram data needs statistical analysis, particularly because the authors state in the text that the data is significant.
We now include in new Figure 2C a bar chart with the data from three independent experiments. As stated in the text, the differences in MT1-MMP signal on EVs were NOT significant (page 12 last paragraph and Figure legend).
Figure 3A. Similar to the previous comment, the FRET data needs some quantitation and statistical analysis to support the contention in the text that the data is significant.
Statistical analyses have now also been performed for FLIM-FRET data in new Figure 3B.
Figure 3C. A whole cell lysate blot of p-ERM would be useful to confirm that p-ERM levels are normal in the mutant transfectants.
We have performed western-blot analyses on whole cell lysates, which revealed no differences in pERM levels upon MT1-MMP wt or mutant transfection or after CRISPR/Cas9 deletion of CD151 (new Figure 4D)
Figure 4. It would be useful to clarify if this data is from stable or transient transfectants.
Data from Figure 4 (Figure 5 in the revised version) relate to transient transfectants in Figure 5B and C and to stable transfectants in Figure 5D, E, F. Quantifications of both autoprocessing or collagenolytic activity show similar results in stable or transient transfections. These details have been added to the text (page 22 under results).
Figure 4A. What is the n number?
n=3 for this graph (now detailed in the Figure legend of new Figure 5A).
Figure 4B and 4E. The data is quantified but some statistical analyses of at least n=3 would be helpful, particularly as there appear to be differences in wild-type ratios between the two panels.
Quantitation of these data has been now provided in new Figure 5F for wt and RRH563AAA mutant in both MCF7 and MCF7-CRISPR/Cas9 CD151 cells.
Figure 4C. What are the relative expression levels of wild-type versus mutant MT1-MMP and does the expression level affect the degree of gelatin degradation? It is importance to establish that this key data is not merely a result of different expression levels.
This question is not easily answered, since gelatin degradation is measured for each individual cell and total MT1-MMP expression can be followed by its GFP tag but the amount of MT1-MMP exposed in the membrane can only be measured by flow cytometry after trypsin treatment. We provide for the reviewer a scatter plot in which we have analysed the area of gelatin degradation and plotted it against the mean GFP fluorescent signal in the ventral confocal plane of the cell. As it can be observed in the plot, there is no clear direct relation between both parameters, supporting a more complex regulation of MT1-MMP that would also involve clustering regulation, autoproteolysis, etc. However, although the lack of correlation impairs further analysis in this way, the plot supports that the RRH563AAA mutant degrades gelatin more efficiently at lower expression levels.
Figure S1. This would seem suitable as a main figure. In panel A, only one negative control histogram is shown, but there are two different cell types; it would be better to have both negatives. In panel B, there is a labelling error: CD9 should be CD151.
Data in Figure S1 is now shown in new Figure 4. We now show both negative controls in the flow cytometry data (completely overlapping). Typo has been corrected, thanks.

Reviewer 2 Report
In this study, the authors aimed at exploring the putative connection of MT1-MMP with the cortical actin cytoskeleton via ERM linkers. They hypothesized that these positively charged clusters of MT1-MMP cytosolic region could enable the interaction with ERM (ezrin, radixin, moesin) proteins and may have an impact on MT1-MMP subcellular distribution and activity. The issue is interesting. However, the previous study indicated that Moesin may regulate cell motility through its interactions with MT1-MMP and E-cadherin/p120-catenin adhesion complex and cytoplasmic expression of Moesin correlates with nodal metastasis and poor prognosis of oral cancer (Li, et al. 2015). In contrast, this study showed that reduced binding action between MT1-MMP and moesin led to increased MT1-MMP gelatinolytic activity. What caused this difference? I have some recommendations for this article. And the active form of MT1-MMP (fig. 4b) had no different levels among these conditions.
In figure 1b, the represented microscopic images of the bar graph on the right side were not completely shown. MT1MMP-RRH563AAA predominantly presented in the intracellular compartment (fig 1b) and more expression in cell surface than wild type (fig 4a). And they were a similar amount in EV (fig 2c). However, in fig 4b, the protein level of MT1MMP seemed to be lower than the others. These differences should be discussed in the text. In fig 4e, RRH”567”AAA or “563”? The others were not shown (under CRISPR/Cas9 CD151). Were there no represented images of EEA-1 and LAMP-1 bar graphs? RRH”576”AAA or “563” (fig 2a)? Have also no represented microscopic images of the bar graph in fig. 2b? The image from MT1MMP-RRH GFP under CRISPR/Casp9 CD151 showed obvious expression of p-ERM. It was not consistent with the band of p-ERM assayed by western blot.Li, Y. Y., C. X. Zhou, and Y. Gao. 2015, Moesin regulates the motility of oral cancer cells via MT1-MMP and E-cadherin/p120-catenin adhesion complex. Oral Oncol 51(10):935-43.
Author Response
Reviewer 2.
Comments and Suggestions for Authors
In this study, the authors aimed at exploring the putative connection of MT1-MMP with the cortical actin cytoskeleton via ERM linkers. They hypothesized that these positively charged clusters of MT1-MMP cytosolic region could enable the interaction with ERM (ezrin, radixin, moesin) proteins and may have an impact on MT1-MMP subcellular distribution and activity. The issue is interesting. However, the previous study indicated that Moesin may regulate cell motility through its interactions with MT1-MMP and E-cadherin/p120-catenin adhesion complex and cytoplasmic expression of Moesin correlates with nodal metastasis and poor prognosis of oral cancer (Li, et al. 2015). In contrast, this study showed that reduced binding action between MT1-MMP and moesin led to increased MT1-MMP gelatinolytic activity. What caused this difference? I have some recommendations for this article. And the active form of MT1-MMP (fig. 4b) had no different levels among these conditions.
Thanks for your comments. The manuscript by Li et al is now quoted under the introduction (last paragraph, ref 56). However, as now discussed on page 27 of the revised manuscript, in that work no direct molecular interaction between MT1-MMP and ERMs was analyzed. In addition, reduction in MT1-MMP total levels was observed upon siRNA silencing of moesin; which may have several indirect effects on other signalling cascades besides actin-cytoskeletal connections.
In figure 1b, the represented microscopic images of the bar graph on the right side were not completely shown.
We now provide in new Figure 1B representative images of the four MT1-MMP mutants quantitated in the graph.
MT1MMP-RRH563AAA predominantly presented in the intracellular compartment (fig 1b) and more expression in cell surface than wild type (fig 4a). And they were a similar amount in EV (fig 2c). However, in fig 4b, the protein level of MT1MMP seemed to be lower than the others. These differences should be discussed in the text.
Overall MT1-MMP levels were as expected very heterogeneous in transient transfectants as revealed by GFP signal, and this may explain the lower signal in Figure 4b (now Figure 5B), in which we only focused on the ratio between processed and non-processed forms. When possible, individual cell analyses were performed. For example, in now Figure 5A, the membrane expression levels were analysed either after gating cells in the intermediately expressing cell on the GFP channel or on the whole population with equal results. As shown above to reviewer 1, no direct relation could be extracted between GFP expression levels and gelatinolytic activity. Thus, the variability in expression levels of the mutants in the different experiments cannot account for the change in activity we demonstrate.
In fig 4e, RRH”567”AAA or “563”? The others were not shown (under CRISPR/Cas9 CD151).
Were there no represented images of EEA-1 and LAMP-1 bar graphs? RRH”576”AAA or “563” (fig 2a)? Have also no represented microscopic images of the bar graph in fig. 2b?
We include now two new supplementary Figures with the whole panels of mutants and stainings that are quantitated in Figure 2A and now Figure 4C. In any case, these supplementary Figures correspond to representative images and many more fields were analysed for the quantitative data.
We have corrected the typos in the Figures, thanks.
The image from MT1MMP-RRH GFP under CRISPR/Casp9 CD151 showed obvious expression of p-ERM. It was not consistent with the band of p-ERM assayed by western blot.
The western-blot in now Figure 4D shows a coimmunoprecipitation assay. In this assay cell lysates were immunoprecipitated with anti-GFP polyclonal antibodies and the associated pERM revealed by WB. The band does not correspond to the total expression of pERM (which does not change upon expression of the different MT1-MMP mutants or deletion of CD151) but to the portion of it that associates to MT1-MMP (only to the wt and not to the RRH563AAA mutant). We now also include in the revised version western-blot analyses on whole cell lysates, which revealed no differences in pERM levels upon MT1-MMP wt or mutant transfection or after CRISPR/Cas9 deletion of CD151 (new Figure 4D)
Round 2
Reviewer 1 Report
I thank the authors for their responses and I am happy with the majority of them. I just have one minor point concerning Figure 1. I appreciate the inclusion of statistics and acknowledge that the data are very clear and striking. I’m sorry to be pedantic, but strictly speaking, I think that multiple t tests should not be used because of the potential for type 1 errors. Instead an ANOVA should probably be used on the entire dataset, followed by multiple pairwise comparisons, assuming that the initial ANOVA showed significance. I believe that the rest of the statistical analyses throughout the paper are fine.
Author Response
Thanks for the positive appreciation. We have replaced the statistical analyses in Figure 1A for the ANOVA test requested.